# Sustained Long-Term Results with Minimal Reintervention Rates in Patients with Frozen Elephant Trunk and Acute Aortic Syndromes

**DOI:** 10.3390/medsci13020052

**Published:** 2025-05-01

**Authors:** Filippos-Paschalis Rorris, Christos F. Pitros, Constantine N. Antonopoulos, Konstantinos Papakonstantinou, Lydia Kokotsaki, Pantelis Tsipas, Ilias Gissis, John Kokotsakis

**Affiliations:** 1Department of Thoracic and Cardiovascular Surgery, Evangelismos General Hospital, 10676 Athens, Greece; 21st Department of Vascular Surgery, Attikon University Hospital, National and Kapodistrian University of Athens, 75000 Athens, Greece; 3Department of Cardiovascular Surgery, 401 General Military Hospital, 11527 Athens, Greece

**Keywords:** frozen elephant trunk, acute aortic syndromes, aortic arch

## Abstract

The Frozen Elephant Trunk (FET) technique is indicated in acute aortic syndromes with arch involvement and malperfusion of tissues. We sought to report on long-term outcomes of FET in emergent cases of acute aortic syndromes. **Methods:** Twenty-three adult patients were referred to our department for surgical management of acute aortic syndromes and underwent aortic arch replacement using the FET technique between November 2010 and January 2022. The primary outcome was long-term survival. Secondary outcomes were the 30-day mortality rate and the incidence of neurologic complications, i.e., stroke and spinal cord ischemia. **Results:** The mean patient age was 57.1 (±12.5) years, and the majority (20 patients, 87%) were male. The most common indication was Stanford type A acute aortic dissection (aTAAD) in 17 (74%) patients, followed by non-A non-B dissection in 2 (8.7%) patients, penetrating aortic ulcer (PAU) of the aortic arch in 2 (8.7%) patients, type A intramural hematoma (IMH) in 1 (4.3%) patient and blunt thoracic aortic injury of the aortic arch in 1 (4.3%) patient. Kaplan–Meier survival analysis revealed a 73% survival at 12 months, which persisted up to 11 years of follow-up. **Conclusion:** The FET technique provides a reliable solution for surgical management of patients with acute aortic syndromes. Excellent, sustained long-term results can be achieved.

## 1. Introduction

The expanding indications for aortic arch replacement using the Frozen Elephant Trunk (FET) technique as well as device evolution mandate a continuous report of outcomes of such patients. Currently, the FET technique is indicated in acute aortic syndromes in the latest European guidelines [1]. Although the FET operation is associated with significant morbidity and mortality, it often offers an excellent option for diseases of the aortic arch and descending thoracic aorta, which is evident by the high rates of downstream re-intervention-free survival in large series [2].

We sought to report on our single institutional experience using the aforementioned technique in patients with acute aortic syndromes operated on in an urgent setting.

## 2. Materials and Methods

We retrospectively extracted data from the department’s database in a single tertiary center on patients who underwent the FET procedure for acute aortic syndromes between November 2010 and January 2022. The Evangelismos Athens General Hospital’s board review and ethics committee approved the study (approval number 9-11-2022/380), and the necessity for patient consent was waived given the retrospective nature of the study. Adult patients who were operated on urgently using the FET procedure in acute aortic syndromes (Stanford type A and type B aortic dissections, intramural hematoma, penetrating aortic ulcer, and blunt thoracic aortic injury) were included. All patients were operated on by the same surgeon (JK). Device choice was based on preoperative computed tomography angiography (CTA) scans as well as device availability. Patients who underwent the FET operation on an elective basis were excluded. The primary objective was long-term survival. Secondary objectives were the 30-day mortality rate, the incidence of neurologic complications, i.e., stroke and spinal cord ischemia and the reintervention rate. Tissue malperfusion (splanchnic, renal, or limb malperfusion) was detected clinically and with CTA imaging preoperatively. Two authors (FPR and JK) were responsible for querying about the patients’ status by contacting them individually. The surgical technique used in our department is reported briefly below and in more detail in our previous publication [3].

### 2.1. Statistical Analysis

Continuous variables were expressed as mean ± standard deviation (SD) values, while categorical variables were presented as raw numbers and percentages (%). Kaplan–Meier analysis was used to estimate the survival of patients after the FET procedure based on long-term follow-up data.

### 2.2. Surgical Technique

During the FET operation, three arterial lines (bi-radial, left femoral) are placed in order to achieve optimal perfusion monitoring, and continuous cerebral monitoring by means of transcutaneous cerebral oximetry (INVOS™ 5100C Medtronic, Minneapolis, MN, USA) was used. Median sternotomy was used in all cases. After complete heparinization, cardiopulmonary bypass (CPB) was instituted via arterial cannulation of the right axillary artery through an 8 mm Dacron graft along standard venous cannulation of the right atrium. Antegrade cold crystalloid cardioplegia (Custodiol 25 mL/kg) was administered after cross-clamping the ascending aorta to achieve cardiac arrest. Once the target bladder temperature of 26 °C was reached, CPB was arrested. The dissection plane and the extent of aortic wall excision depend on the aortic condition and are planned preoperatively. For patients in whom the island technique was used, the distal ascending aorta and aortic arch just proximal to the origin of the left subclavian artery (LSA) were excised. An island with the origins of the innominate artery (IA) and LCCA was created while the origin of LSA was ligated. Unilateral selective antegrade cerebral perfusion was initiated through the right axillary artery after snaring the IA and LCCA. In cases of aortic dissection, an angioscope is used to introduce a guidewire in the true lumen of the descending thoracic aorta. The hybrid stent-graft system was then introduced antegradely through the open aortic arch in the descending thoracic aorta in an over-the-wire fashion and released with a pull-back system. The cuff of the hybrid prosthesis is anastomosed to the targeted aortic zone stump with a 3.0 polypropylene suture and externally reinforced with a Teflon strip. Immediately after the distal anastomosis, the side branch was used for lower body perfusion after clamping the vascular part of the hybrid graft proximally. The island was then implanted into the Dacron part of the vascular part and the proximal anastomosis was carried out. Consecutively, the right axillary arterial perfusion is rescinded, and the perfusion side branch is used exclusively for perfusion.

## 3. Results

Since the FET operation was first introduced in our department in 2007, a total of 67 patients have been operated on using that technique. However, the FET operation was first implemented in a patient with acute aortic syndrome in November 2010, and thus the study period was set to begin at that time. Thus, between November 2010 and January 2022, 23 patients with acute aortic syndromes who underwent the FET procedure were identified through the hospital’s records and included in this report. Follow-up was available for all patients.

The mean patient age was 57.1 (±12.5) years, and the majority (20 patients—87%) were male. The most common indication was Stanford type A acute aortic dissection (aTAAD) in 17 (74%) patients, non-A non-B dissection in 2 (8.7%) patients, penetrating aortic ulcer (PAU) of the aortic arch in 2 (8.7%) patients, type A intramural hematoma (IMH) in 1 patient and blunt thoracic aortic injury of the aortic arch in 1 patient. Devices used were the E-Vita Open (Artivion, Kennesaw, GA, USA) in 1 patient, E-Vita Open Plus (Artivion, Kennesaw, GA, USA) in 14 (60.7%) patients, Thoraflex Hybrid Device (Vascutek Ltd., Inchinnan, UK) in 3 (13%) patients and the E-Vita Open Neo (Artivion, Kennesaw, GA, USA) in 5 (21.7%) patients. The FET procedure was elected over other types of surgical aortic dissection repair due to (i) malperfusion, (ii) an entry tear in the aortic arch or proximal descending aorta, (iii) multiple entry tears in the ascending aorta and aortic arch, and (iv) an aneurysmal aortic arch and proximal descending aorta.

Patient baseline characteristics as well as periprocedural data are listed in Table 1. Seventeen patients remain alive at the last follow-up. Kaplan–Meier survival analysis revealed a 73% survival at 12 months, which persisted up to 11 years of follow-up (Figure 1). There were two intraoperative deaths due to low cardiac output syndrome, and the overall 30-day mortality rate was 22% (five patients). Three patients (13%) had a stroke postoperatively, but none were left with permanent neurologic disability. Two patients (8.7%) developed delayed, transient spinal cord ischemia, which did not cause permanent neurologic disability. Additionally, three patients (13%) required reintervention with a completion thoracic endovascular aortic repair (TEVAR) due to type Ib endoleak.

## 4. Discussion

We reported on long-term outcomes of patients who underwent total aortic arch replacement with the FET technique for acute aortic syndromes in an urgent setting. Follow-up was available for up to 11 years. The majority of the patients (82.3%) included in this report had suffered an acute aortic dissection, and almost half of our cohort (11 patients) had clinical signs of either splanchnic, renal, or limb malperfusion. Notably, none of our patients died postoperatively due to malperfusion. This finding is in line with international guidelines, which recommend the use of FET for the treatment of patients suffering an acute aortic dissection associated with malperfusion signs and symptoms.

Furthermore, we have provided long-term survival and follow-up data of our patient cohort. As seen in the Kaplan–Meier survival curve, there is an initial drop in survival during the first six months. After that, however, the curve parallels the x-axis until the end of the 11-year follow-up which suggests that there were no deaths during that time. After the initial early mortality, we did not observe a change in the survival of patients. Furthermore, we have shown a sustained 73% survival, which persists up to the end of follow-up.

The FET procedure is associated with serious morbidity and significant complications [4]. Perioperative complications such as prolonged ventilation, which is attributed to acute respiratory distress syndrome (ARDS), and the need for temporary renal dialysis for acute kidney injury were noted in our patients, but, in our experience, patients are usually able to overcome such complications given their relatively young age. Neurologic insults such as stroke and spinal cord ischemia are always a concern during aortic dissection repair and FET. Two of our patients had a cerebrovascular accident, but none were left with a permanent neurologic disability. Similarly, two patients exhibited late spinal cord ischemia, which was treated promptly, and none of them were left with permanent disability. In our institution, we have implemented the COPS protocol [5] for the early management of spinal cord ischemia.

In the majority of our patients, the stent graft segment of the hybrid device was able to “land” in the non-aneurysmal aorta and thus seal distally. In cases of aneurysmal descending aorta, the FET device cannot provide a distal seal and type Ib endoleak develops. The reintervention rate in our cohort was minimal, and it was limited to TEVAR in three patients due to type Ib endoleak. The stent graft part of the device offers an excellent landing zone for TEVAR. Typically, we treat the endoleak during the index hospitalization for acute aortic syndrome. Novel devices have tried to mitigate the problem of aneurysmal distal aorta by providing grafts of different sizes between the vascular graft and the stent graft of the device and by tapering between the two, as well as increasing lengths of the stent graft part, reaching 180 mm [3].

Proximalization of the distal anastomosis has been made possible with the launch of the new device E-Vita Open Neo. The device allows for distal anastomosis in zones 0 and 1, in addition to the “classic” zone 2 or 3 anastomosis. Proximalization of the distal anastomosis in the FET technique is associated with better survival and less incidence of neurologic complications [4]. The latest device referenced above has several additional advantages over its competitors, one of which is the tapering of the Dacron and stent graft part of the device as well as different available lengths of the stent graft part. This allows for optimal graft selection and true individualization for each patient. Although we predominantly used the E-Vita devices, mostly due to availability in our center, the Thoraflex hybrid device is an excellent option for the FET procedure.

## 5. Conclusions

The FET procedure in acute aortic cases is associated with increased perioperative morbidity and mortality. However, minimal reintervention rate and sustained long-term results place the FET technique at the top of the pyramid as a surgical method of choice in acute aortic syndromes.

## Figures and Tables

**Figure 1 medsci-13-00052-f001:**
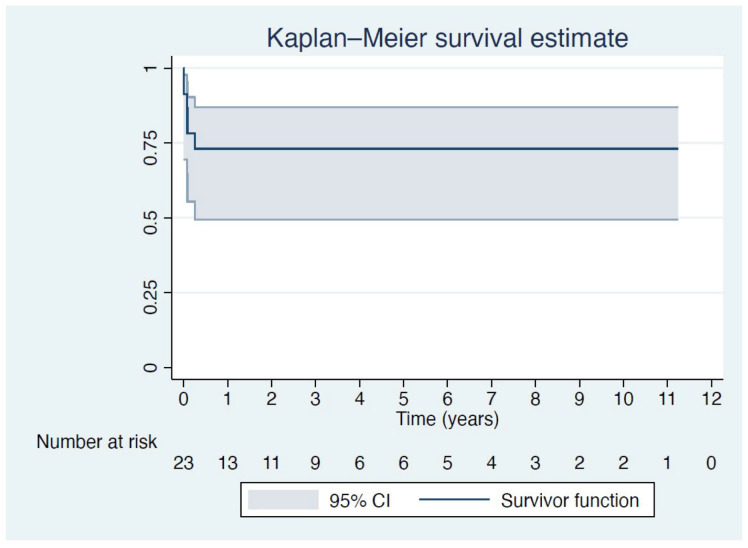
Kaplan-Meier survival analysis.

**Table 1 medsci-13-00052-t001:** Baseline patient characteristics and periprocedural data.

Variable	N (%)
Age	57 (±12.5)
Male gender	20 (87%)
Hypertension	19 (82.3%)
Diabetes	1 (4.3%)
COPD	5 (21.7%)
Renal dysfunction	3 (13%)
Marfan syndrome	1 (4.3%)
Aortic regurgitation (III–IV)	1 (4.3%)
Hemopericardium	5 (21.7%)
Malperfusion, splachnic	3 (13%)
Malperfusion, renal	5 (21.7%)
Malperfusion, upper and lower limbs	3 (13%)
**Periprocedural data**	
Cannulation site	
Right axillary artery	20 (87%)
Left common carotid artery	2 (8.7%)
Right femoral artery	1 (4.3%)
CPB time (min)	216.42 (98–326)
Myocardial ischemia time (min)	148.82 (75–240)
Peripheral ischemia time (min)	51.63 (32–73)
SACP (min)	89.76 (55–120)
Body temperature (°C)	24.15 (21–26)
**Postoperative data**	
Causes of death	
Low cardiac output syndrome	2 (8.7%)
Multiorgan failure	2 (8.7%)
Sepsis	1 (4.3%)
Complications	
Ventilation > 96 h	6 (26%)
Temporary dialysis	5 (21.7%)
CVA	2 (8.7%)
SCI	2 (8.7%)
Resternotomy (tamponade)	1 (4.3%)

Abbreviations: COPD, chronic obstructive pulmonary disorder; CPB, cardiopulmonary bypass; SACP, selective antegrade cerebral perfusion; CVA, cerebrovascular accident; SCI, spinal cord ischemia.

## Data Availability

The data that support the findings of this study are available upon request from the corresponding author.

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
