# Peer review of "Sustained Long-Term Results with Minimal Reintervention Rates in Patients with Frozen Elephant Trunk and Acute Aortic Syndromes"

_medsci, 2025, doi:10.3390/medsci13020052_

Round 1
Reviewer 1 Report
Comments and Suggestions for Authors
It is an interesting manuscript on an important issue in the therapy of acute aortic syndrome. However, the presented data shows - in a quite small cohort - a high mortality and morbidity. As such, based on the small cohort, any clear statistical conclusion cannot be drawn.
I addition, I do not see the clear mid-/long-term benefit of this approach, especially if you have such a high initial mortality and morbidity. It would als be interesting how many different surgeons did the surgeries, and what decided which device was used.
Nevertheless, I would like to congratulate the authors for their interesting and nice manuscript, which - with some revisions - could be suggested for publication in the journal.
Comments on the Quality of English Languagenone
Author Response
Thank you for your kind comments. We have now added additional information in the Materials and Methods section clarifying who performed the operations as well as the device of choice.
Reviewer 2 Report
Comments and Suggestions for Authors
Your manuscript describes a challenging population undergoing aortic surgery with notable complexity and complications. However, it requires more comprehensive detail to ensure scientific rigor. The methodology and inclusion/exclusion criteria need further clarification—particularly regarding whether the cohort was elective or emergency, and how malperfusion was diagnosed and addressed. In addition, baseline echocardiographic and imaging data, definitions for key endpoints (e.g., 30-day or in-hospital mortality), and methods of neurological monitoring should be specified to contextualize the reported outcomes. A clearer account of the rationale for each cannulation site and a discussion of how your complication rates compare to existing literature would enhance the clinical relevance. Finally, given the small sample size, a more transparent approach to statistical analysis and acknowledgment of potential confounders will bolster the study’s credibility.
Author Response
Thank you for your important comments.
The cohort was emergency cases, and we have tried to make that more clear in our methodology section.
We have now included a separate section describing our surgical technique with methods of neurologic monitoring perioperatively, cannulation site chosen, etc.
Reviewer 3 Report
Comments and Suggestions for Authors
Manuscript ‚Sustained long-term results with minimal reintervention rates in patients with Frozen Elephant Trunk and acute aortic syndromes‘, Rorris et al.
Thank you for the opportunity to review the interesting manuscript, I have a few questions and comments.
- I congratulate the authors on the good results of FET implantation in acute aortic syndromes in a relatively low volume setting with only two cases per year on average.
- Line 2, what was the reintervention rate? I did not find anything in the text alluding to reintervention rates. May be lines 86/87 indicate a reintervention. But this is very marginally described. When one of the main focuses of the study is a low reintervention rate, this has to be elaborated more thoroughly.
- Line 27, ‚permanent solution‘ goes into the same direction. What ist the evidence given in the manuscript, that the FET technique provides a permanent solution? The low intervention rate (which is not really described, see above)? The 73% survival at 12 months? Please explain.
- Line 81, in the files I had access to, I could not find a figure 1.
- Line 93 and following, the description of properative malperfusion fits better into the Results chapter.
- Line 98, I could not find a Kaplan-Meier curve.
- Lines 110 and following, description of neurologic events also fits better into the Results chapter.
- Lines 137-139, what evidence is given in the text concerning superiority of FET implantation as opposed to other surgical techniques?
Author Response
Thank you for your comments.
All reinterventions were completed TEVAR operations. We have tried to make this clearer in lines 117/118 of our revised manuscript.
In addition, we have revised our methodology section as well as added a surgical technique section.
The word “permanent” has been replaced.
We apologize for not being able to view our Kaplan analysis. We hope that will be amended in the revision.
Round 2
Reviewer 1 Report
Comments and Suggestions for Authors
The authors made mostly adequate corrections.
Reviewer 2 Report
Comments and Suggestions for Authors
Ok to proceed